# Recalibrating the Why and Whom of Animal Models in Parkinson Disease: A Clinician’s Perspective

**DOI:** 10.3390/brainsci14020151

**Published:** 2024-01-31

**Authors:** Andrea Sturchio, Emily M. Rocha, Marcelo A. Kauffman, Luca Marsili, Abhimanyu Mahajan, Ameya A. Saraf, Joaquin A. Vizcarra, Ziyuan Guo, Alberto J. Espay

**Affiliations:** 1James J. and Joan A. Gardner Family Center for Parkinson’s Disease and Movement Disorders, Department of Neurology, University of Cincinnati, Cincinnati, OH 45219, USA; andrea.sturchio@gmail.com (A.S.); luca.marsili@uc.edu (L.M.); sarafae@mail.uc.edu (A.A.S.); 2Pittsburgh Institute for Neurodegenerative Diseases, Department of Neurology, University of Pittsburgh, Pittsburgh, PA 15213, USA; rocha@pitt.edu; 3Consultorio y Laboratorio de Neurogenética, Centro Universitario de Neurología José María Ramos Mejía, Buenos Aires C1221ADC, Argentina; marcelokauffman@gmail.com; 4Department of Neurology, University of Pennsylvania Perelman School of Medicine, Philadelphia, PA 15213, USA; joaquin.vizcarrap@gmail.com; 5Center for Stem Cell and Organoid Medicine (CuSTOM), Division of Developmental Biology, Cincinnati Children’s Hospital, Department of Pediatrics, College of Medicine, University of Cincinnati, Cincinnati, OH 45229, USA; ziyuan.guo@cchmc.org

**Keywords:** animal model, Parkinson’s disease, loss-of-function hypothesis, neurodegeneration, organoids

## Abstract

Animal models have been used to gain pathophysiologic insights into Parkinson’s disease (PD) and aid in the translational efforts of interventions with therapeutic potential in human clinical trials. However, no disease-modifying therapy for PD has successfully emerged from model predictions. These translational disappointments warrant a reappraisal of the types of preclinical questions asked of animal models. Besides the limitations of experimental designs, the one-size convergence and oversimplification yielded by a model cannot recapitulate the molecular diversity within and between PD patients. Here, we compare the strengths and pitfalls of different models, review the discrepancies between animal and human data on similar pathologic and molecular mechanisms, assess the potential of organoids as novel modeling tools, and evaluate the types of questions for which models can guide and misguide. We propose that animal models may be of greatest utility in the evaluation of molecular mechanisms, neural pathways, drug toxicity, and safety but can be unreliable or misleading when used to generate pathophysiologic hypotheses or predict therapeutic efficacy for compounds with potential neuroprotective effects in humans. To enhance the translational disease-modification potential, the modeling must reflect the biology not of a diseased population but of subtypes of diseased humans to distinguish *What* data are relevant and to *Whom*.

## 1. Introduction

Research involving animal models aims at translating pre-clinical scientific discoveries and hypotheses into human clinical trials. While animal models have been fundamental in several fields of medicine, including but not limited to general pharmacology and oncology, they have been unsuccessful in fostering the development of disease-modifying therapies in Parkinson’s disease (PD) [1]. By design, animal models can only replicate a small fraction of the complexity and diversity of human physiology. However, the data generated in animal models are assumed to be generalizable to the large spectrum of human disease. This intrinsic limitation is recognized as a major obstacle to the translational potential of experimental therapeutics and of any pathophysiological mechanism being examined [2].

The major goal of animal modeling is to understand the pathophysiology of a disease. However, it is important to acknowledge that (1) most diseases (including PD) have subtypes and (2) existing animal models can reliably model some aspects of a disease, not all its subtypes. Therefore, any conclusions made using animal models should always acknowledge the heterogeneity of the disease it models. As the underlying pathological mechanism(s) responsible for clinical symptoms vary for groups of affected individuals, animal models need to be selected according to the PD subtype they represent. We suggest that animal models should be used to understand a specific biological mechanism, such as molecular interactions or receptor–drug interactions as applied to the subset of affected individuals for which they were created, and independently of the final behavioral outcome. This approach may result in the discovery of cellular mechanisms associated with such a disease subtype, which will in turn serve to identify the appropriate target population for a clinical trial, a requirement for the achievement of precision medicine [3].

Here, we review the pros and cons of each of three main types of animal model, highlighting their “one-size-fits-all” approach and their emphasis on the “toxic proteinopathy” (gain of function) model. We argue that the misuse of animal models in predicting treatment responses generalizable across the biological spectrum of PD may be the single most important gap preventing translational successes in disease modification. We highlight new approaches offered by the organoid models and assess the types of questions for which each model can provide help and for which they can misguide in translating putative neuroprotective interventions into humans.

### 1.1. Clinicopathological Fallacy

The clinicopathologic model is based on attributing convergence and causality to post-mortem pathological findings of aggregated α-synuclein into Lewy bodies and Lewy neurites (collectively, Lewy pathology (LP)). However, LP is neither sufficient nor necessary for neurodegeneration or PD symptoms to appear [4]. For instance, LP can be found in many elderly subjects without signs of parkinsonism [4] and most often coexists with other co-pathologies [5,6,7]. Furthermore, many genetic forms of parkinsonism are not invariably associated with LP, such as *LRRK2-* and *PRKN*-PD [7]. The low sensitivity and specificity of LP argue against its role as a disease determinant and, in turn, as a target for biomarker-discovery and disease-modifying efforts. If a wide range of biological alterations are responsible for common clinical and pathologic phenotypes, the prevailing proteinopathy-based framework for PD may represent a common consequence rather than a finding of pathogenic relevance in each affected person [8]. In fact, in a retrospective study of 1647 autopsied individuals, many pathologies were common in both PD and in normal people, with up to seven pathologies occurring in one hundred and sixty-one combinations, yielding a match between pathology and diagnosis of only between 19% to 45% [9].

For the purposes of precision medicine, the main conceptual advantage of animal models is to provide the means of evaluating the mechanisms and toxicity of potential neuroprotective interventions. However, missing from these preclinical efforts is the development of bioassays linking those mechanisms to the population biologically suitable to benefit. No refinements in clinical trial design, sensitivity of endpoints, or target engagement can overcome the fundamental mismatch in clinical trials between the *What* (intervention) and the *Who* (recipient), which can only be achieved through the assay-based screening of candidates biologically suitable to benefit from the intervention of interest [10].

### 1.2. Pros and Cons of Available Animal Models

The most common animal models used in PD include: (1) neurotoxic models; (2) genetic models; and (3) viral vector-based models (Table 1).

*Neurotoxic models* are based on dopaminergic neuron degeneration induced by the administration of agents with robust selective toxicity to neurons in the substantia nigra (SN), giving rise to motor and behavioral symptoms with neurodegeneration without LP [11]. Discovered from a synthetic “designer” heroin used by six patients evaluated by Dr. Bill Langston in the early 1980s, the mitochondrial complex-I inhibitor, 1-methyl-4phenyl-1,2,3,6-tetrahydropyridine (MPTP), became the most popular neurotoxic model [12], inspiring over 9000 publications since 1983 (Figure 1). However, these models developed a parkinsonism that today would not be recognized as PD: rapid-onset with early postural impairment, waxy flexibility, frontal release signs, and apraxia of eyelid opening among other atypical features. Because of the rapid timeline of injury, the ensuing neurodegenerative phase may not be reliable for pathophysiological studies. In addition, some protocols for the MPTP models cause phenotypic suppression of dopaminergic neurons rather than neuronal loss [13]. Other neurotoxic models such as rotenone or paraquat also have limitations [13]. For example, paraquat models exhibit LP formation but the neuronal loss is much lower than in MPTP; moreover, paraquat does not cross the blood–brain barrier and it is associated with systemic damage [13]. Rotenone exerts varying lesion extension and sensitivity in different rat strains [13]. A recent paper proposed that one of the reasons for the failure is that it is impossible to fully replicate the results of preclinical models in clinical trials because the progression in humans is extremely slow (around 70 dopaminergic neurons per day from a pool of 800,000/1,000,000) compared to animal models [14]. Interestingly, they suggested testing a specific pathway activation, such as KEAP1/NRF2, which might have a potential neuroprotective effect by increasing the expression of DT-diaphorase and glutathione transferase M2-2 that can prevent neurotoxicity from free neuromelanin production [14].

Although neurotoxin models are easy to handle, these models may better reflect a late stage of PD when the chronic damage is already present. However, in their brief initial phase (when toxicity is not yet associated with neuronal loss), they may also be interpreted as a “prodromal” disease phase [15]. These models have been commonly used to test neuroprotective interventions in disease-modifying PD trials; while the effect size of each of the experimental interventions which underwent translation was promising, there was no translation of such efficacy in human clinical trials (Figure 2) [16,17,18,19,20,21,22,23,24,25,26]. Thus, the neurotoxic models may be more appropriate for evaluating the overall safety of interventions and the symptomatic effects of dopamine-enhancing (e.g., symptomatic) treatments but inappropriate for estimating the disease-modifying effect of putative neuroprotective treatments, at least as applicable to the majority of PD subtypes.

*Genetic models* have been argued to more closely reflect PD progression beyond human genetic subtypes. However, these are not without substantial concerns.

Highly penetrant autosomal dominant or recessive genetic variations account for only about 2% of PD cases, with the percentage increasing to 5–10% for genetic variations associated with increased PD risk, such as LRRK2 and GBA mutations [27]. Also, the genetic expression and splicing events can differ between human and animal models [28]. Critically, different variants within the same gene (see LRRK2) are associated with several different types of parkinsonian syndromes [29].

With the exception of the extremely rare familial subtypes with 100% penetrance, all others, including GBA and LRRK2, have a relatively low penetrance.

Transgenic animal models do not replicate the wide spectrum of PD progression, demonstrating, in some cases, a very early-onset of a pathological phenotype even during embryogenesis [30], quite different from what has been observed in humans.

The anatomical distribution of LP is very different among animal models. For instance, although many transgenic lines show typical markers of LP [31,32,33,34,35,36], the distribution of these aggregates is different from the ones observed in PD patients. The SN in these models, invariably affected in PD patients, is often spared or minimally involved [31,32]. The distribution of aggregated α-synuclein could reflect a loss of the endogenous soluble precursor in a specific neuronal region (see “Models of α-synuclein”).

Because of these pathological differences, the parkinsonian motor phenotype results in neurotransmitter deficiencies other than dopamine and could arise from the involvement of other areas of the central nervous system. For example, some animal models show an involvement of the spinal cord [32,33,34,35,36,37] and this could be associated with pyramidal features to a greater extent than parkinsonism. 

*Viral overexpression of α-synuclein* is associated with similar shortcomings. In these models, neurodegeneration is highly variable but with generally faster progression compared to human PD [38,39,40,41]. For instance, at earlier stages of PD, there is a ~90% preservation of SN neurons [42] suggesting that early alterations are predominantly functional over structural, with important implications for therapeutic development. 

In addition, the behavioral tests in animal models are often misleading. Many such tests are based on assessments that could reflect not only motor alterations. In fact, most behavioral evaluations require a learning process and are not only strictly associated with dopamine-dependent motor skills [28]. This learning process can happen before the experimental lesion of dopaminergic neurons, potentially adding an important confounder in the analysis [30]. The timing of the lesions needs to be carefully selected according to the specific research question. Moreover, in animal models, tremor and rigidity are difficult to evaluate compared to bradykinesia [30]. Finally, we need to consider the variability in environmental factors; for instance, paradoxically, some PD animal models show hyperactivity, rather than bradykinesia, probably due to anxiety-like behaviors [43,44,45].

## 2. Models for What?

The main shortcoming in all animal models is not that the underlying molecular mechanisms remain unknown, but that any replication of their effects has unclear parallels among humans with PD. While animal models based on a specific genetic alteration would be expected to reflect the biological alterations responsible for disease expression in the humans affected by those genetic variants, the expectation has largely been that any genetic model is a model of “sporadic” PD—implicitly representing most or everyone affected. This assumption, in many cases, may have justified neglecting a major step in the translational value of animal model research: bioassay development for use in clinical trials. Any compound that shows promise in rodents by preventing behavioral deficits and neuropathologies requires a clinical bioassay that can identify that particular subtype of PD.

The evaluation of a drug’s toxicity is the earliest necessary step in any therapeutic development and is required for regulatory approvals [46]. But a non-toxic drug based on animal data can be irreparably toxic in humans and vice versa. In fact, if tested in animal models, some available drugs would have never been approved. For example, BIA-102474-101, a drug with anti-anxiety properties designed for PD led to deep brain hemorrhage and necrosis and one death at dosages much lower than those used in nonhuman primates and dogs [47]. On the other hand, the widely available drug, paracetamol, is known to be toxic in dogs and cats [48]. The lack of animal testing, however, was responsible for the “Elixir Sulfanilamide disaster of 1937” [49], a sulfonamide antibiotic which was responsible for many deaths. In summary, despite some animal–human discrepancies, the assessment of toxicity in animals remains critical for limiting the application in humans of drugs with unfavorable risk/benefit trade-offs. 

## 3. Models for Whom?

The prevailing hypothesis for PD holds that the pathophysiological mechanism responsible for PD is the accumulation of the aggregated form of α-synuclein into crossed-β pleated sheets, namely LP. Many other pathogenetic mechanisms have been associated with the onset of PD and different animal models have been developed to study each possible mechanism. These animal models assume that the mechanism(s) responsible for any observed pathology applies to all PD patients, rather than as an instrument of alterations suitable for testing putative disease-modifying treatments in the corresponding small subtype of PD.

*Examples of models of neuroinflammation.* One of the commonly used models to study neuroinflammation is the lipopolysaccharides (LPSs), which bind to toll-like receptor (TLR) 4 on the microglia and macrophages [50]. LPSs do not cross the blood–brain barrier and, therefore, must be injected directly into the brain—typically the SN or striatum. This approach will result in the extensive degeneration of dopaminergic neurons (up to 80%), eliciting dopamine and motor alterations albeit without consistent PD features [51,52,53,54,55,56]. A single LPS infusion leads to a release of inflammatory molecules, particularly tumor necrosis factor α (TNF-α), interleukin (IL)-6 or IL-1β, nitric oxide (NO), and superoxide [55,57,58,59,60]. While this model can be informative of specific “inflammatory” PD subtypes, it was conceived as supporting the pathogenicity of neuroinflammation in all or most patients affected. The attempts at generalizing the output of neuroinflammation models are hampered by the fact that microglia activation and subsequent inflammation in humans is caused by many factors [61] and LPSs can lead to different pathological processes in animal and humans [62].

*Examples of models of mitochondrial dysfunction.* Both neurotoxic and genetic models have been used to analyze mitochondrial dysfunction. For example, one of the suggested mechanisms of toxicity suggested for the overexpression of wild-type α-synuclein or missense point mutation, such as A53T and E46K, is the mitochondrial fragmentation through the alteration of the interference with the mitochondria-associated membranes or through the downregulation of the peroxisome proliferator-activated receptor-γ coactivator 1-α [63,64]. *LRRK2* models, instead, interfere with the mitochondrial fission proteins, such as dynamin-related protein 1, mitofusin 1/2, and optic atrophy 1 [65], and impair the proteasomal degradation of Miro, affecting mitophagy [66]. The mechanism of Parkin toxicity mainly involves the degradation of dysfunctional mitochondria [67] and the impairment of mitochondrial biogenesis [68]. None of these models are capable of reflecting the pathology and progression of the full range of sporadic PD with each exhibiting a different clinicopathological phenotype.

*Examples of models of lysosomal dysfunction.* Many genetic data converge on the association of lysosomal dysfunction with PD risk. Genes involved in lysosomal function, such as *ATP13A2*, *TMEM175*, *CTSD*, and *GBA*, have been associated with an increased risk of PD. *ATP13A2* encodes for ATPase pump, a transmembrane lysosomal protein involved in cation and polyamine transport [69]; knock-out *ATP13A2* mice show sensorimotor deficits but with no loss of dopaminergic neurons; moreover, they show features atypical for PD, such as an increase in lipofuscin in the cerebellum and hippocampus and accumulation of insoluble α-synuclein in the hippocampus [70]. Instead, *TMEM175* encodes for the lysosomal potassium channel TMEM175; knock-out animal models show increased α-synuclein pathology in the neurons and accumulation of α-synuclein in the primary hippocampal neurons of rats [71]. *CTSD* encodes for the lysosomal enzyme cathepsin D, which degrades proteins responsible for the degradation of long-lived proteins. Similar to humans with a congenital deficit for cathepsin, *CTSD* knock-out animal models show diffuse α-synuclein inclusions in the deep cortical laminae, superior colliculi, subiculum, thalamus, cerebellum, and white matter tracts but with less atrophy than in humans [72].

*Examples of GBA models.* Inhibition of the *GBA* gene is responsible for a range of clinical and pathological phenotypes. GBA encodes the lysosomal sphingolipid degrading enzyme glucocerebrosidase (GCase) [73]. Homozygous *GBA1* mutations are responsible for Gaucher disease (GD), in which the loss of GCase activity results in the accumulation of glucocerebroside (GlcCer) and glucosylsphingosine (GlcSph) [73]. Different variants are associated with different risks and clinicopathological manifestations; in general, GBA-associated parkinsonism exhibits more pronounced synucleinopathy compared with sporadic PD [74]. The accumulation of GlcCer and GlcSph in those with *GBA* mutations promotes α-synuclein aggregation [75,76,77] and is toxic to the endoplasmic reticulum by causing endoplasmic reticulum-associated protein degradation (ERAD) stress [78,79]. Many GBA1 animal models, however, are discrepant with the corresponding human GBA-associated parkinsonian phenotypes. The L444P mutation is responsible for a severe form of GBA-associated parkinsonism and mice carrying the L444P heterozygote mutation in the murine GBA gene show high levels of α-synuclein, although in the absence of aggregates, dopaminergic degeneration, and signs of inflammation [80]. But for this animal model to manifest a phenotype, it must be associated with an *SNCA* mutation (A53T, A30P), viral overexpression of α-synuclein via an adeno-associated virus, or “facilitated” with MPTP [80,81,82,83]. On the other side, the more common N370S mutation shows a mild phenotype in humans, whereas it is associated with high neonatal mortality in animal models [80,84].

In summary, each animal model may represent a standard measure but only for a small subset of human PD. Given that the development of disease-modifying therapies is dependent on the identification of neuroprotective molecules working through specific mechanisms in certain disease subtypes, models also need to generate a bioassay translatable into clinical trial programs to permit the identification of the subset of PD patients with whom such a mechanism may be active and pathogenic. 

### 3.1. Models of α-Synuclein Accumulation vs. Depletion

The hypothesis that protein accumulation is toxic originated from autopsy observations attributing pathogenicity to what was visible under a microscope (pathology), overlooking that which had become invisible: the soluble, functional, largely monomeric α-synuclein precursor. As a result, most animal models have pursued modeling the gain-of-function (GOF) toxicity while distinctly fewer models have examined the alternative loss-of-function (LOF) toxicity. 

*GOF hypothesis.* The major support for this hypothesis has come from the discovery in humans of genetic mutations in the gene coding for α-synuclein, *SNCA* [85]. Transgenic overexpression of human or wild-type α-synuclein or human missense mutation variants, such as A53T, are the most common genetic animal models. The overexpression of α-synuclein can be enhanced in all neurons (e.g., human platelet-derived growth factor subunit B, mouse thymus cell antigen 1 promoter) or limited to a specific subpopulation of neurons (e.g., tyrosine hydroxylase promoter for catecholaminergic neurons in the SN) [86,87,88,89] according to the type of promoter selected for the study. A major weakness in the direct translation of the GOF hypothesis is the weak clinical correlation with degeneration in humans [90], and in animal models [30]. Also, many of these models do not show SN neuronal loss, whereas others show SN degeneration in the absence of brain α-synuclein aggregates (Table 2) [34,40,91,92,93,94,95,96,97,98,99,100,101,102,103].

Importantly, the toxicity of SNCA mutations may not arise from the “overexpression” of α-synuclein but from the depletion of its inherently unstable and aggregation-prone precursor. In fact, SNCA point mutations or multiplications reduce the soluble α-synuclein pool by facilitating their aggregation through the mechanism of homogenous nucleation, whereby the nucleation barrier or the energy threshold for peptides to aggregate is lowered when their concentration increases [104]. This has been demonstrated in Alzheimer’s disease, where triplication of the APP gene forces the soluble 42 amino acid β-amyloid (Aβ42) peptide to aggregate, reducing its levels in cerebrospinal fluid [105]. The importance of the reduction in endogenous α-synuclein in the presence of a high concentration of α-synuclein is highlighted by the treatment of animal models with prefibrillar forms (PFF) of α-synuclein, which induces the aggregation of endogenous α-synuclein. Toxic effects appear if the endogenous levels of α-synuclein are reduced; in fact, this effect is null in α-synuclein knock-out animal models [106,107], possibly due to the inability to recruit endogenous α-synuclein.

*LOF hypothesis.* The major support for this hypothesis comes from the fact that proteins must be in their soluble state to function normally; misfolding into a crossed-β pleated sheet invariably leads to loss of their normal function. Against this hypothesis, it has been argued that the germline α-synuclein knock-out animal model has shown either no or very mild clinical phenotypes [108,109,110,111,112]. This may be due to the known redundancy in the synuclein family, in the form of β- and γ-synuclein, which are part of the same protein type and share similar functions. In fact, α-synuclein knock-out animal models overexpress β- and γ-synuclein in order to compensate for the absence of α-synuclein [112,113,114]. Moreover, the deletion of α- and β-synuclein is responsible for an increase in γ-synuclein [115], whereas the overexpression of β-synuclein is responsible for the downregulation of α-synuclein [116], further confirming the interaction among and functional redundancy between these molecules. Animal models in which all these three synucleins are silenced have shown neurodegeneration and increases in mortality [117], supporting the importance of their physiologic, monomeric (unaggregated) state. Moreover, a recent study showed that triple knockout (KO) models for the synuclein family show impairment in the endocannabinoid system through a postsynaptic alteration in the SNARE complex, affecting the release of endocannabinoids, which are crucial for neuroplasticity; for instance, the deficiency in endocannabinoids is associated with the onset of PD [118]. Moreover, it has been shown that the overexpression of α-synuclein completely prevents the mortality and neuronal damage in cysteine string protein (CSPα) KO models [119]. Interestingly, some studies have shown that a reduction in α-synuclein in adult models, in which the compensatory redundancy of other families of synucleins may be degraded, is associated with the clinical and pathological signs of PD in the absence of LP (by design) [120,121,122,123], even in non-human primates [124]. In some animal models with minimal effects after adult suppression of endogenous α-synuclein, the reduction in α-synuclein may have been insufficient (only around 30–50%) to facilitate the phenotype [125,126,127,128,129]. Benskey and colleagues have suggested that a reduction of more than 50% of α-synuclein achieved for ≥7 days is necessary [123], although Zharikov and colleagues have shown that a reduction of 70% of α-synuclein was not associated with neurodegeneration [130]. Methodological differences between these studies, such as the type of vector, the shRNA sequences, the vector doses and infusion rates, and the types of promoters may account for some of these differences [130]. In summary, many studies have highlighted the importance of a high (normal) level of endogenous soluble α-synuclein with an acute reduction in mature neurons associated with neuronal damage and, eventually, neuronal death. It has also been shown that as α-synuclein colocalizes with the molecules involved in DNA damage, its reduction affects DNA repair [131].

### 3.2. Alternative Models: Organoids Derived from iPSCs

While genetic animal models reproduce single genetic mutations, brain organoids can be generated from human-induced pluripotent stem cells (iPSCs) obtained from patients with sporadic or inherited PD, capturing the unique genomic background of each individual. This approach facilitates the study of personalized mechanisms and drug responses tailored to each individual. Organoids are self-organizing 3D tissue cultures that mimic the properties of organs [132]. Specifically, guided organoids, able to express specific patterning factors that allow differentiation into specific brain areas, can be particularly useful for understanding the first steps of the pathophysiological aspects of specific subtypes of PD via the use of midbrain organoids. An untapped potential of organoids lies in their capacity to simultaneously screen multiple drugs, offering a promising avenue for research and drug discovery [133], although an interaction of drugs with other organs will still need the use of animal models. Other limitations are still present for this purpose, such as the cellular heterogeneity or difference across protocols with consequent lack of reproducibility and the difficulties with their scalability [134]. These limitations have been discussed by Costamagna and colleagues [134].

Organoids possess distinct brain regional identities and layered architectures, closely resembling the human brain. They exhibit essential features of neural development, including progenitor zones, maturing neurons, and glia, making them a valuable tool for modeling neurodevelopmental aspects, particularly relevant in early-onset genetic cases of PD. Organoids also develop functional neural networks, displaying synchronized electrophysiological patterns reminiscent of preterm human brains. This capacity to model higher-order circuitry enables the study of how different PD genetics may disrupt complex processes. Therefore, the pathological study of patient-derived brain organoids and their disease-related higher brain order dysfunction can not only be benchmarked against clinical pathological data from patients but also offer a unique opportunity to study the causal link between genetics and symptoms, which can eventually close the gap between the bench and clinic. 

By expressing endogenous levels of PD-related proteins, organoids avoid the non-physiological overexpression in genetic models. The intrinsic ability to recapitulate patient-specific phenotypes makes organoids well-suited for stratifying diverse PD subtypes and testing personalized therapies, in contrast to animal models that converge on a single phenotype. Opportunities for further refinement of the tool include the xenografting of human brain organoids into in vivo animal models to increase vascularization and survival time [135], the compartmentalization of brain regions into “assembloids” to better match normal physiology [136], and the generation of midbrain organoids containing microglia [137].

Another potential advance with the transplantation of organoids in animal models is that they may anatomically and functionally integrate with animal models [138]. This could improve the use of animal models in research. However, these models may not serve to evaluate complex mechanisms due to interaction between multiple systems, such as the influence of the microbiome. In summary, organoids may become a desirable alternative to animal models by enabling the evaluation of structures that better reflect normal and pathologic human physiology. 

### 3.3. Alternative Models: Genetically Humanized Mice

A new CRISPR–Cas9 technique allows the rewriting of stretches of DNA in mouse embryonic stem cells to humanize mice models. This “stepwise swapping strategy” (mSwAP-In) technique was used to recapitulate human-specific expression and splicing patterns of several human-specific ACE2 transcripts, which expressed in distinct tissues on the hACE2 mice model (helpful to study COVID-19) [139]. The mSwAP-In method can be used to humanize any gene or set of genes in mice, enabling the rapid production of personalized animal models without time-consuming breeding. This technique is under further refinement given the risks of deleting and overwriting genes that are essential in mouse embryonic stem, which could potentially delete crucial functional regulatory elements of distant genes [139].

### 3.4. The “Mechanoidal Phase”: Implications for Modeling the Complexity of PD

In his conception of the “mechanoidal phase”, Stephen Wolfram—a multifaceted computational scientist and physicist—described biological systems as exhibiting a higher-level molecular organization that is neither simply ordered nor completely random. He likens this phase to the complex, machine-like organization of elements in systems like microprocessors, where the detailed arrangement of components defines functional outputs. According to Wolfram, the mechanoidal phase depends on the precise choreography of molecular interactions, in contrast to something like a gas where assuming “molecular chaos” is often sufficient. While computationally irreducible, the mechanoidal phase contains pockets of reducibility in the form of identifiable structures and mechanisms. Wolfram’s mechanoidal phase highlights the limitations of using convergent animal models to understand the complexity of PD. Their forced convergence on singular dopaminergic phenotypes reflects our limited observational capabilities as “analog biological computers”, rather than an accurate reflection of the diverse molecular mechanisms dysregulating neuronal function across PD subtypes. As Wolfram notes, the detailed histories and interactions of individual elements are critical in complex systems like biology. But in using reductionist animal models, we ignore the unique temporal trajectories and non-linear pathways within each PD patient’s vulnerable neurons. Though we perceive orderly structures as akin to ‘wires and switches’, the functional choreography underlying neural computation remains obscure. Animal models provide islands of computational reducibility, but PD remains an irreducibly complex illness with intricate genetic and environmental interactions defining the neuronal fate. While animal models offer pockets of insight, we must develop new tools like brain organoids and human neuronal cultures that better capture the diverse genomic backgrounds and neuronal pathophysiologies of PD subtypes. 

## 4. Conclusions

The consistent translational failure of disease-modifying therapies into humans suggests that the testing of pathophysiologic hypotheses or the prediction of therapeutic efficacy of interventions with putative neuroprotective effect in humans may not be the best application of animal models. Individuals with different expressions of disease cannot be properly evaluated using a global model of disease. While the most immediate utility of animal models is in the evaluation of drug toxicity and safety and the understanding of underlying mechanisms, future iterations based on brain organoids may afford the testing of neuroprotective interventions in biologically ascertained recipients. To enhance the output of research efforts in disease modification, modeling must reflect the biology not of a diseased population but of a given subtype of humans with that disease in order to match the *What* to the *Who*, the basic principle of precision medicine.

## 5. Disclosures

A Sturchio cofounded REGAIN Therapeutics and is co-inventor of the patent “Compositions and methods for treatment and/or prophylaxis of proteinopathies”. 

EM Rocha has nothing to disclose.

M Kauffman has received grant support from the Ministry of Science and Technology from Argentina and is an active employee of CONICET (Argentinean’s national science committee).

L Marsili has received honoraria from the International Association of Parkinsonism and Related Disorders (IAPRD) Society for social media and web support, and personal compensation as a consultant/scientific advisory board member for Acadia. Dr. Marsili has received a grant (collaborative research agreement) from the International Parkinson and Movement Disorders Society for the MDS-UTRS Validation Program (Role: PI), a Non-Profit.

A Mahajan has received grant funding from the DMRF, the Parkinson’s Foundation, and the Sunflower Parkinson’s disease foundation outside of the submitted work. He reports no conflicts of interest.

AA Saraf has nothing to disclose.

JA Vizcarra serves on the editorial board of the journal *Neurology*.

Z Guo has received grant support from the NIH and the Local Initiative for Excellence Foundation.

AJ Espay has received grant support from the NIH and the Michael J Fox Foundation; personal compensation as a consultant/scientific advisory board member for Neuroderm, Amneal, Acadia, Avion Pharmaceuticals, Acorda, Kyowa Kirin, Supernus (formerly, USWorldMeds), and Herantis Pharma; personal honoraria for speakership for Avion, Amneal, and Supernus; and publishing royalties from Lippincott Williams & Wilkins, Cambridge University Press, and Springer. He cofounded REGAIN Therapeutics and is co-inventor of the patent “Compositions and methods for treatment and/or prophylaxis of proteinopathies”. He serves on the editorial boards of the *Journal of Parkinson’s Disease*, the *Journal of Alzheimer’s Disease*, the *European Journal of Neurology*, *Movement Disorders Clinical Practice*, and *JAMA Neurology.*

## Figures and Tables

**Figure 1 brainsci-14-00151-f001:**
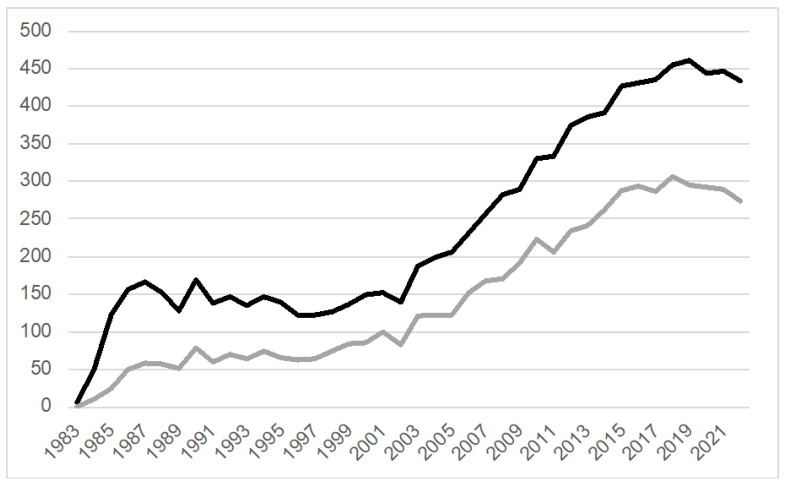
**MPTP-related publications (PubMed, 1983–2022).** A total of 9608 MPTP-related research projects (black line) have been published since the first report in 1983. Of those, 5768 publications (gray line) are associated with the keywords “treatment” or “therapy”. Search terms black line: (MPTP OR 1-Methyl-4-phenyl-1,2,3,6-tetrahydropyridine OR (“1-Methyl-4-phenyl-1,2,3,6-tetrahydropyridine”[Mesh]) OR “MPTP Poisoning”[Mesh]) NOT (review OR “Review” [Publication Type]); gray line: ((MPTP OR 1-Methyl-4-phenyl-1,2,3,6-tetrahydropyridine OR (“1-Methyl-4-phenyl-1,2,3,6-tetrahydropyridine”[Mesh]) OR “MPTP Poisoning”[Mesh]) NOT (review OR “Review” [Publication Type])) AND (treatment OR therapy).

**Figure 2 brainsci-14-00151-f002:**
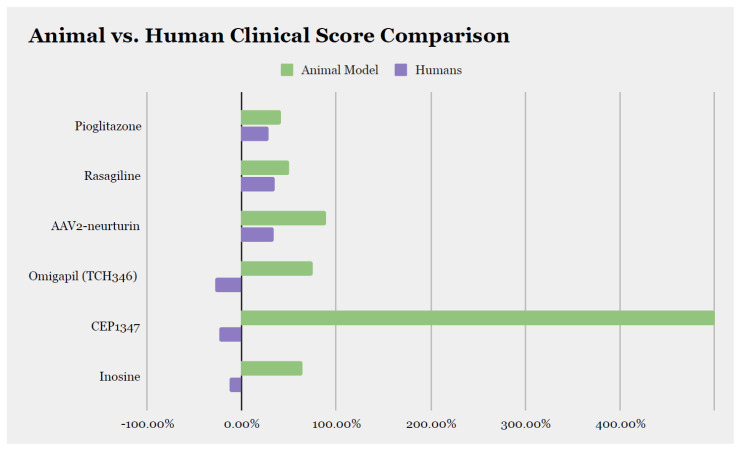
**Comparison of the effect size of disease-modifying therapies between animal model data and clinical trial data.** We calculated the percentage of improvement or worsening at the last assessment compared to baseline and included molecules for which the score was available in both animal models and humans. AAV2: adeno-associated virus type-2.

**Table 1 brainsci-14-00151-t001:** Features, benefits, and limitations of different types of animal models.

Models	Features	Benefits	Limitation
Neurotoxic (MPTP, 6-OHDA)	Absence of α-synuclein aggregates Rapid and diffuse dopaminergic neurodegeneration Presence of motor deficits	Discovery of symptomatic drugs Used in mice and Non-human primates	No pathological analogy with human PD No prediction of success in translational efforts for DMT Lack of prodromal stages
Genetic models	Presence of some motor deficits Slower onset of symptoms Some do not exhibit α-synuclein aggregation or dopaminergic neuronal loss	Evaluates α-synuclein aggregation process Study changes in α-synuclein aggregation	Mutations often affect the physiological development of the animal. Presence of α-synuclein pathology in other areas (e.g., spinal cord) that could interfere with the evaluation of motor symptoms Inconsistent clinicopathological phenotype among different models.
Viral transfection of α-synuclein	Presence of α-synuclein aggregates Moderate dopaminergic neurodegeneration Presence of motor deficits	Evaluates α-synuclein aggregation process	Potential vector toxicity and trigger for α-synuclein aggregation Damage limited to site of injection

MPTP: 1-methyl-4-phenyl-1,2,3,6-tetrahydropyridine; 6-OHDA: 6-hydroxydopamine; PD: Parkinson’s disease; DMT: disease-modifying therapies.

**Table 2 brainsci-14-00151-t002:** Transgenic and viral vector-based mice/rat models of α-synuclein.

Type of Promoter/Virus	Type of Mutation	α-Synuclein Brain Aggregates	SN Cell Loss
SNCA	hWT hA53T	− +	− +
Th	hWT hA53T; hA53T + A30P	+ −	− +
Thy1	hWT	+	−
Prnp	hA53T hA30P	+ −	− −
PDGFB	hWT	+	−
AAV	hWT; hA53T	+	+

AAV, adeno-associated virus; hA30P, human A30P mutant; hA53T, human A53T mutant; hWT, human wild type; PDGFB, gene encoding platelet–derived growth factor subunit B; Prnp, gene encoding prion protein; SN, substantia nigra; SNCA, gene encoding α-synuclein; Th, gene encoding tyrosine hydroxylase; Thy1, gene encoding thymus cell antigen 1; −, absent; +, present.

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
