# Peer review of "Recalibrating the Why and Whom of Animal Models in Parkinson Disease: A Clinician’s Perspective"

_brainsci, 2024, doi:10.3390/brainsci14020151_

Round 1
Reviewer 1 Report
Comments and Suggestions for Authors
This manuscript points to a truly fundamental problem in the design of new drugs that can modify the course of Parkinson's disease both by delaying the evolution of the disease and by slowing the progression of the disease. The authors postulate that the cause of the failure of clinical studies is related to preclinical animal models that do not represent what happens in the disease.
The authors conclude that the use of a global disease model does not represent what happens in individuals with different expression of the disease. Although there are differences in the expression of the disease, all patients finally converge in the loss of dopamine neurons that contain neuromelanin.
The authors hypothesize that interactions between brain organoids could be fundamental to achieving future drugs that have ultimate success in clinical studies. However, neither cell cultures nor brain organoid cultures will be able to replace the interactions of the organs of a human being where drug metabolism in the liver and intestine play key role drugs metabolism that finally exert a therapeutic effect on the Parkinsons’s disease brain.
The authors do not take into account a recent publication (Neural Regen Res. 2024 Mar;19(3):529-535. doi: 10.4103/1673-5374.380878.) that proposes that the failure of clinical studies based on neurotoxins is due to the degenerative process is extremely slow, taking years before the appearance of motor symptoms and the progression of the disease. Following this argument, this would imply that it is impossible to replicate the successful results in preclinical models of Parkinson's since the progress of the disease is so slow that it prevents positive effects from being observed in a patient with Parkinson's disease. This model of "Single-neuron neurodegeneration as a degenerative model for Parkinson's disease" suggests that the neurotoxin that triggers the degeneration of dopaminergic neurons that contain neuromelanin would be aminochrome. But this work suggests that no preclinical animal model where a neurotoxin, even aminochrome, is injected, cannot replicate this single-neuron degenerative model since all neurotoxins have an expansive effect. This publication suggests to test molecules that activate the KEAP1/NRF2 pathway that induces the expression of 2 enzymes that prevent the neurotoxic effects of aminochrome that would trigger the degenerative process of Parkinson's disease.
Author Response
Comment #1: This manuscript points to a truly fundamental problem in the design of new drugs that can modify the course of Parkinson's disease both by delaying the evolution of the disease and by slowing the progression of the disease. The authors postulate that the cause of the failure of clinical studies is related to preclinical animal models that do not represent what happens in the disease. The authors conclude that the use of a global disease model does not represent what happens in individuals with different expression of the disease. Although there are differences in the expression of the disease, all patients finally converge in the loss of dopamine neurons that contain neuromelanin. The authors hypothesize that interactions between brain organoids could be fundamental to achieving future drugs that have ultimate success in clinical studies. However, neither cell cultures nor brain organoid cultures will be able to replace the interactions of the organs of a human being where drug metabolism in the liver and intestine play key role drugs metabolism that finally exert a therapeutic effect on the Parkinsons’s disease brain.
Answer #1: We thank the Reviewer for the nice comment. We also agree that neither cell cultures nor brain organoids can fully recapitulate Parkinson’s disease. We added some more limitations of this model in the paragraph “Alternative models: organoids derived from iPSCs.”
Comment #2:The authors do not take into account a recent publication (Neural Regen Res. 2024 Mar;19(3):529-535. doi: 10.4103/1673-5374.380878.) that proposes that the failure of clinical studies based on neurotoxins is due to the degenerative process is extremely slow, taking years before the appearance of motor symptoms and the progression of the disease. Following this argument, this would imply that it is impossible to replicate the successful results in preclinical models of Parkinson's since the progress of the disease is so slow that it prevents positive effects from being observed in a patient with Parkinson's disease. This model of "Single-neuron neurodegeneration as a degenerative model for Parkinson's disease" suggests that the neurotoxin that triggers the degeneration of dopaminergic neurons that contain neuromelanin would be aminochrome. But this work suggests that no preclinical animal model where a neurotoxin, even aminochrome, is injected, cannot replicate this single-neuron degenerative model since all neurotoxins have an expansive effect. This publication suggests to test molecules that activate the KEAP1/NRF2 pathway that induces the expression of 2 enzymes that prevent the neurotoxic effects of aminochrome that would trigger the degenerative process of Parkinson's disease.
Answer #2: We thank the Reviewer for sharing this important paper that we missed to add. We have now added this important paper (page 3; lines 115-120).
Reviewer 2 Report
Comments and Suggestions for Authors
Review of a manuscript “Recalibrating the Why and Whom of Animal Models in Parkin[1]son’s Disease: A Clinician’s Perspective” by Andrea Sturchio and coauthors submitted to “Brain Sciences”.
Parkinson’s disease is the second after Alzheimer’s disease neurodegenerative disorder for which there are no yet disease-modifying therapies. Animal models may be helpful for translating pre-clinical scientific discoveries and developments into human clinical trials. The authors review the advantages and drawbacks of several types of animal models including the organoids models. This is an important field of biomedical science, and the manuscript will be interesting for the readership of the “Brain Sciences”.
The following corrections and additions should be made.
Page 2. “This approach may result in the discovery of cellular mechanisms associated with such subtypes
of disease, which will in turn serve to identify the appropriate target population for a clinical trial, a requirement for the achievement of precision medicine.3”
The authors should be more specific trying to explain how many subtypes of Parkinson’s disease they mean. Is it realistic to develop an animal model for each of these subtypes?
Page 3. The authors should add a reference in the following sentence: “Discovered from a synthetic “designer” heroin used by six patients evaluated by Dr. Bill Langston in the early 1980s, the mitochondrial complex-I inhibitor, 1-methyl-4phe[1]nyl-1,2,3,6-tetrahydropyridine (MPTP) became the most popular neurotoxic model [reference: Emamzadeh et al., Parkinson’s disease: Biomarkers, Treatment, and Risk Factors. Frontiers in Neuroscience, Neurodegeneration, 12, 61230, 2018. https://doi.org/10.3389/fnins.2018.00612], inspiring over 9,000 publications since (Figure 1).
Figure 1. MPTP-related publications (Pubmed, 1983-2022). There is discrepancy between the data on the figure and in the Figure legend. On the Figure the last year included is 2021, on figure legend 2022.
Overall 1. In addition to the existence of different subtypes of Parkinson’s disease in humans, there is a heterogeneity of animal models. Even various mice strains may give different responses to the tested drug, not to say about differences for various animal species. How this ambiguity should be taken into consideration? This issue should be discussed.
Overall 2. The authors are critically consider many weakness of animal models, but they add more information about possibilities to improve these drawbacks.
In general, this is an interesting manuscript containing new important information which needs some clarification and improvements.
Author Response
Rev2:
Comment #1: Parkinson’s disease is the second after Alzheimer’s disease neurodegenerative disorder for which there are no yet disease-modifying therapies. Animal models may be helpful for translating pre-clinical scientific discoveries and developments into human clinical trials. The authors review the advantages and drawbacks of several types of animal models including the organoids models. This is an important field of biomedical science, and the manuscript will be interesting for the readership of the “Brain Sciences”.
Answer #1: We thank the Reviewer for the nice comment.
Comment #2: The following corrections and additions should be made.
Page 2. “This approach may result in the discovery of cellular mechanisms associated with such subtypes
of disease, which will in turn serve to identify the appropriate target population for a clinical trial, a requirement for the achievement of precision medicine.3”
The authors should be more specific trying to explain how many subtypes of Parkinson’s disease they mean. Is it realistic to develop an animal model for each of these subtypes?
Answer #2: We thank the Reviewer for the nice comment. We do not know how many subtypes of PD exist. However, we think that an animal model could be used if the final outcome is to analyze a specific mechanism. We have added this on page 2 lines 58-61: “We suggest that animal models should be used to understand a specific biological mechanism, such as molecular interactions or receptor-drug interactions as applied to the subset of affected individuals for which it was created, and independently of the final behavioral outcome.”
Comment #3: Page 3. The authors should add a reference in the following sentence: “Discovered from a synthetic “designer” heroin used by six patients evaluated by Dr. Bill Langston in the early 1980s, the mitochondrial complex-I inhibitor, 1-methyl-4phe[1]nyl-1,2,3,6-tetrahydropyridine (MPTP) became the most popular neurotoxic model [reference: Emamzadeh et al., Parkinson’s disease: Biomarkers, Treatment, and Risk Factors. Frontiers in Neuroscience, Neurodegeneration, 12, 61230, 2018. https://doi.org/10.3389/fnins.2018.00612], inspiring over 9,000 publications since (Figure 1).
Answer #3: We have now added the reference.
Comment #4: Figure 1. MPTP-related publications (Pubmed, 1983-2022). There is discrepancy between the data on the figure and in the Figure legend. On the Figure the last year included is 2021, on figure legend 2022.
Answer #4: We apologize for the confusion. The problem is that the first study starts in 1983 and the legends goes 2 years at the time. So the graph is correct but the legend stops at 2021.
Comment #5: Overall 1. In addition to the existence of different subtypes of Parkinson’s disease in humans, there is a heterogeneity of animal models. Even various mice strains may give different responses to the tested drug, not to say about differences for various animal species. How this ambiguity should be taken into consideration? This issue should be discussed.
Answer #5: We thank the Reviewer for this comment. We think that this problem can be tackled in the manner we discussed in comment 2, with the corresponding change made to the manuscript. The choice of the model and strain should be based on the mechanism and type of question that needs to be studied.
Comment #6: Overall 2. The authors critically consider many weaknesses of animal models, but they add more information about possibilities to improve these drawbacks. In general, this is an interesting manuscript containing new important information which needs some clarification and improvements.
Answer #6: We thank the Reviewer, and we hope the improvements satisfied her/his concerns.
Reviewer 3 Report
Comments and Suggestions for Authors
Sturchio et al. discuss the limitations or ‘translational disappointments’ originating from use of animal models of Parkinson disease. While the authors highlight the disappointments or rather failures of translational research with respect to animal models, they promote new models, such as the use of brain-derived organoids, as a potential alternative. I commend the authors that the review was written relatively well, but as presented, it is not publishable. My comments are below:
1. For completion, the authors should discuss and critically consider the cumulative research using iPSC-derived organoid models in the context of Parkinson disease. What are the present or future limitations of organoid models? For example: What are the present and future limitations of transplanted organoid models if the ‘animal component’ is (re)added as a model? What are the implications of interactions of genes and environment (e.g. microbiome) in the context of organoid research? How can drug interactions with other organs be assessed in a specific organoid model? Science and research are not rosy and successful all the time. To give an unbiased picture, a holistic criticism, and discussion of the present research into the use of ‘Parkinson’ organoids is necessary. For example, refer to Int J Mol Sci. 2021 Mar 6;22(5):2659. doi: 10.3390/ijms22052659 and related articles or refer to the primary research and discuss critically.
2. In the abstract, rephrase ‘… therapeutic efficacy of interventions with putative neuroprotective effect in humans …’ to remove redundancies.
3. On page 1, last line, ‘they’ can refer to ‘humans’ and ‘animal models’, though it is not unintended. Please rephrase to avoid ambiguity.
4. Second paragraph, page 2: use singular after ‘each’.
5. In Subtitle ‘Clinico-pathological’, remove the hyphen.
6. Page 3, first paragraph: do you mean ‘These models’ instead of ‘These patients’? No line numbers are present, so reviewing becomes difficult.
7. Page 6, para 4: write ‘point mutations’ instead of ‘points mutation’.
8. Page 7, second line: use ‘show increased’.
9. Page 10, first paragraph: use ‘obscure’ as an adjective, not a past participle.
10. Throughout the manuscript, revise the possessive ‘Parkinson’s’ because despite being the commonly used and commonly accepted “misnomers”, the eponymous terms “Parkinson’s Disease” and “Alzheimer’s Disease” are logically incorrect. Historically, the diseases were discovered by Charles Parkinson and Alois Alzheimer, respectively; the diseases were not “their own” diseases. Because of the eponymous convention, using the possessive form (apostrophe plus “s” or genitive “s”) is wrong but has been perpetuated in the English Scientific literature by our great peers. Many though have avoided it. The Australian Manual of Scientific Style and The Chicago Manual of Style also advise against the use of the possessive form. I suggest taking their editorial advice and applying it throughout the text. See references 3, 4, 15, 19, 23, 78, and 80 cited in your text.
11. Revise the list of references thoroughly as the list seems to be presented to the readers perfunctorily. For example, the numbering is wrong when 7 and 8 are corrected. The latter two refer to the same source. Revise the titles and the author names to match the sources exactly. For example, the Latin α appears as ‘alpha’ and ‘a’. In reference 72, what does ‘a_ects’ mean? In reference 81, add a missing space in the title. The same for reference 93.
Comments on the Quality of English LanguageSee comments above.
Author Response
Rev3:
Sturchio et al. discuss the limitations or ‘translational disappointments’ originating from use of animal models of Parkinson disease. While the authors highlight the disappointments or rather failures of translational research with respect to animal models, they promote new models, such as the use of brain-derived organoids, as a potential alternative. I commend the authors that the review was written relatively well, but as presented, it is not publishable. My comments are below:
Comment #1: For completion, the authors should discuss and critically consider the cumulative research using iPSC-derived organoid models in the context of Parkinson disease. What are the present or future limitations of organoid models? For example: What are the present and future limitations of transplanted organoid models if the ‘animal component’ is (re)added as a model? What are the implications of interactions of genes and environment (e.g. microbiome) in the context of organoid research? How can drug interactions with other organs be assessed in a specific organoid model? Science and research are not rosy and successful all the time. To give an unbiased picture, a holistic criticism, and discussion of the present research into the use of ‘Parkinson’ organoids is necessary. For example, refer to Int J Mol Sci. 2021 Mar 6;22(5):2659. doi: 10.3390/ijms22052659 and related articles or refer to the primary research and discuss critically.
Answer #1: We thank the Reviewer for this comment. We have added this important component in the paragraph “Alternative models: organoids derived from iPSCs.”
Comment #2: In the abstract, rephrase ‘… therapeutic efficacy of interventions with putative neuroprotective effect in humans …’ to remove redundancies
Answer #2: We have revised this part.
Comment #3: On page 1, last line, ‘they’ can refer to ‘humans’ and ‘animal models’, though it is not unintended. Please rephrase to avoid ambiguity.
Answer #3: We thank the Reviewer for pointing out this. We have now revised the sentence.
Comment #4: Second paragraph, page 2: use singular after ‘each’.
Answer #4: We have modified this part
Comment #5: In Subtitle ‘Clinico-pathological’, remove the hyphen.
Answer #5: We have now removed the hyphen.
Comment #6: Page 3, first paragraph: do you mean ‘These models’ instead of ‘These patients’? No line numbers are present, so reviewing becomes difficult.
Answer #6: We apologize for the confusion. We have now corrected this part.
Comment #7: Page 6, para 4: write ‘point mutations’ instead of ‘points mutation’.
Answer #7: We have now corrected it.
Comment #8: Page 7, second line: use ‘show increased’.
Answer #8: We have modified this part according to the Reviewer’s suggestion.
Comment #9: Page 10, first paragraph: use ‘obscure’ as an adjective, not a past participle.
Answer #9: We have now corrected it.
Comment #10: Throughout the manuscript, revise the possessive ‘Parkinson’s’ because despite being the commonly used and commonly accepted “misnomers”, the eponymous terms “Parkinson’s Disease” and “Alzheimer’s Disease” are logically incorrect. Historically, the diseases were discovered by Charles Parkinson and Alois Alzheimer, respectively; the diseases were not “their own” diseases. Because of the eponymous convention, using the possessive form (apostrophe plus “s” or genitive “s”) is wrong but has been perpetuated in the English Scientific literature by our great peers. Many though have avoided it. The Australian Manual of Scientific Style and The Chicago Manual of Style also advise against the use of the possessive form. I suggest taking their editorial advice and applying it throughout the text. See references 3, 4, 15, 19, 23, 78, and 80 cited in your text.
Answer #10: We thank the Reviewer for this important comment. We have now eliminated the apostrophe in “Parkinson disease” throughout the manuscript.
Comment #11: Revise the list of references thoroughly as the list seems to be presented to the readers perfunctorily. For example, the numbering is wrong when 7 and 8 are corrected. The latter two refer to the same source. Revise the titles and the author names to match the sources exactly. For example, the Latin α appears as ‘alpha’ and ‘a’. In reference 72, what does ‘a_ects’ mean? In reference 81, add a missing space in the title. The same for reference 93.
Answer #11: We thank the Reviewer for this comment. We have now corrected this.
Round 2
Reviewer 1 Report
Comments and Suggestions for Authors
Recommendation: Accept in present form